# Effects of Hypoxia Stress on Survival, Antioxidant and Anaerobic Metabolic Enzymes, and Related Gene Expression of Red Swamp Crayfish *Procambarus clarkii*

**DOI:** 10.3390/biology13010033

**Published:** 2024-01-06

**Authors:** Qinghui Zeng, Mingzhong Luo, Lirong Qin, Chao Guo, Jiashou Liu, Tanglin Zhang, Guangpeng Feng, Wei Li

**Affiliations:** 1State Key Laboratory of Freshwater Ecology and Biotechnology, Institute of Hydrobiology, Chinese Academy of Sciences, Wuhan 430072, China; 13137617716@163.com (Q.Z.); qinlirong@ihb.ac.cn (L.Q.); guochao@ihb.ac.cn (C.G.); jsliu@ihb.ac.cn (J.L.); tlzhang@ihb.ac.cn (T.Z.); 2College of Animal Science and Technology, Yangtze University, Jingzhou 434025, China; kklmz413@yangtzeu.edu.cn; 3Jiangxi Institute for Fisheries Sciences, Poyang Lake Fisheries Research Centre of Jiangxi Province, Nanchang 330039, China; fgp7711@163.com

**Keywords:** environmental stress, oxygen consumption rate, enzyme activity, stress genes, crayfish

## Abstract

**Simple Summary:**

In order to investigate the effects of low oxygen stress on the survival rate, antioxidant, anaerobic metabolic enzymes, and related genes of crayfish during aquaculture. This experiment conducted a 24-h low oxygen stress experiment on juvenile and subadult crayfish, and the results showed that the higher the degree of low oxygen stress, the higher the antioxidant and anaerobic metabolic enzyme activity and gene expression levels of juvenile crayfish and subadult crayfish. Compared with subadult crayfish, the antioxidant system of crayfish larvae is more sensitive to hypoxic environments.

**Abstract:**

The red swamp crayfish *Procambarus clarkii* is the most reared shrimp in China, but it is often affected by hypoxia stress in the process of seedling culture and adult crayfish culture. The oxygen consumption rate and asphyxiation point of juvenile crayfish (1.17 ± 0.03 g) and subadult crayfish (11.68 ± 0.11 g) at different temperatures (20, 22, 24, 26, and 28 °C) were studied. The survival, glycolysis, and expression of antioxidant genes were compared under 24 h acute hypoxia stress (1, 2, and 3 mg/L) and normal dissolved oxygen (7.5 mg/L). The results showed that the oxygen consumption rate and asphyxiation point of juvenile and subadult crayfish increased with increasing temperatures (20–28 °C). At the same temperature, the oxygen consumption rate and asphyxiation point of juvenile crayfish were significantly higher than those of subadult crayfish (*p* < 0.05). Within 24 h, the three hypoxia stress environments did not lead to the death of crayfish, indicating that *P. clarkii* has a strong ability to adapt to hypoxia. Hypoxia stress significantly affected the activities of antioxidant and anaerobic metabolic enzymes and gene expression in juvenile and subadult crayfish. The activities of the superoxide dismutase (SOD), catalase (CAT), and lactate dehydrogenase (LDH) and the content of lactic acid (LD) in the hepatopancreas of juvenile and subadult crayfish in the hypoxia stress groups increased significantly. The expression levels of *SOD* mRNA, *CAT* mRNA, *Hsp*70 mRNA, and *crustin 4* mRNA in the hepatopancreas of juvenile and subadult crayfish in the hypoxia stress groups were significantly higher than those in the control group (*p* < 0.05), and the higher the degree of hypoxia stress, the higher the expression of each gene. The results showed that the antioxidant system of juvenile crayfish was more sensitive to hypoxia environments, and hypoxia stress resulted in increased stress levels in juvenile crayfish and subadult crayfish.

## 1. Introduction

Red swamp crayfish (*Procambarus clarkii*) originated in the south-central United States and northeastern Mexico and were introduced into China from Japan in the 1930s [1]. Due to their rapid growth and high nutritional and commercial value, artificial culture of *P. clarkii* developed rapidly from the 1980s to 2020s, and their distribution rapidly expanded to more than 20 provinces in China, becoming the crustacean with the highest yield in China and the second highest yield in the world [2,3]. In 2021, the total area of crayfish culture in China reached 1.73 million ha, with an output of 2.63 million tons, accounting for 8.27% of China’s total output of freshwater aquaculture. The output value of the aquaculture industry was approximately 11.76 billion USD [4]. At present, the main mode of *Procambarus clarkii* (*P. clarkii*) culture in China is integrated rich crayfish. However, problems such as high mortality and slow growth often occur in the process of crayfish culture in rice fields, affecting the economic benefits to farmers and restricting the sustainable development of crayfish culture.

There are a variety of factors affecting the mortality and growth rate of aquatic animals, including biological and abiotic factors. Biological factors mainly include predators, competitive species, and food resources. Abiotic factors include feed feeding, stocking density, and culture environment, among which the culture environment is considered to be one of the important factors affecting the survival and growth of crustaceans [5,6]. Previous studies have shown that the deterioration of the quality of the aquaculture environment poses a great challenge to the survival of crustaceans, including temperature stress [7], pH stress [8], ammonia stress [9], hypoxia stress [10], and other factors that often lead to disease and death in crustaceans.

In an integrated rice and crayfish cultivation system, the use of fertilizer in the process of rice planting leads to the enrichment of nutrients in the sediment. The decomposition of straw returned to the field after a rice harvest also leads to an increase in nutrients in the water and a rapid change in the physical and chemical indices of water quality. Therefore, the phenomena of low pH, high nitrite, and low dissolved oxygen often occur during the culture of *P. clarkii* in rice fields [6,11]. France found that the tolerance of juvenile *Orconectes virilis* to pH changes was lower than that of adult individuals, which led to the death of juvenile crayfish when the pH was lower than 5.5 [12]. Excessive nitrite nitrogen will reduce the oxygen carrying capacity of blood, resulting in hypoxia and even pathological changes in tissues and organs [6]. Nitrite stress can decrease the activity of SOD and CAT, decrease antioxidant capacity, and affect the immune function of *P. clarkii* [13]. Hypoxia stress can significantly affect the growth, metabolism, and antioxidant systems of *Litopenaeus vannamei* (*L. vannamei*) [14], *Penaeus vannamei* [15], and *Macrobrachium rosenbergii* [16]. However, there are few studies on the adaptability to hypoxia of *P. clarkii* and the effects of hypoxia stress on the immune antioxidant system and related gene expression.

Crustaceans mainly show innate immune responses to hypoxia stress, including physical defense and humoral immunity [17,18]. Previous studies have shown that hypoxia stress can increase the content of superoxide anions in the hepatopancreas and the activity of SOD in the gills and hepatopancreas of *L. vannamei* [14,19]. Moreover, the expression levels of *LvGAPDH* mRNA and glyceraldehyde-3-phosphate dehydrogenase in gill tissue and phosphofructokinase and fructose-1,6-diphosphatase in the hepatopancreas of *L. vannamei* were significantly increased [14]. Short-term (≤7 days) hypoxia stress can increase the expression of HIF-1a and enhance the antioxidant response in the hepatopancreas and midgut of *L. vannamei*, while long-term (≥14 days) cyclic hypoxia stress can destroy the cellular antioxidant mechanism in the hepatopancreas and midgut, inhibit the antioxidant response, and aggravate cell apoptosis and histopathological lesions in the hepatopancreas and midgut [20]. Under hypoxic conditions, the intestinal flora and mucosal morphology of *Macrobrachium nipponense* (*M. nipponense*) changed, and the intestinal immune enzyme activity decreased [21]. The enzyme activity related to glucose anaerobic glycolysis increased, the mRNA expression level of aerobic respiration decreased, and the normal metabolic activity changed from aerobic respiration to anaerobic respiration, which disturbed the energy metabolism of *M. nipponense* and affected the defense regulation of the antioxidant system under a hypoxia environment [13,16,22]. These studies show that hypoxia stress can significantly affect the immune and antioxidant capacity of crustaceans and ultimately affect their normal survival and growth.

Thus, it was hypothesized that the activities of antioxidant enzymes, those of anaerobic respiratory enzymes, and the heat stress gene expression levels of *P. clarkii* increased under anoxic environments, and there were differences between juvenile and subadult *P. clarkii*. In this study, to explore the hypoxia adaptability of *P. clarkii* and to understand the changes in the antioxidant system and anaerobic metabolic enzymes under hypoxic conditions, we measured the respiratory rate and asphyxiation point of juvenile and subadult *P. clarkii* at different temperatures. The expression levels of antioxidant enzymes, anaerobic metabolic enzymes, and antioxidant and stress genes in the hepatopancreas of juvenile and subadult *P. clarkii* under different hypoxia stresses were studied. The differences in the antioxidant systems and anaerobic metabolic enzymes between juvenile crayfish and subadult crayfish under the same hypoxic stress were compared. The results of this investigation help to reveal the physiological and biochemical reactions of *P. clarkii* under hypoxic conditions, providing a theoretical basis for improving the survival rate of *P. clarkii* in farming.

## 2. Materials and Methods

### 2.1. Experimental Animals and Ethics

Cultured juvenile crayfish (1.05 ± 0.05 g) and subadult crayfish (11.21 ± 0.15 g) were obtained from the crayfish breeding farm of Yangtze University in Jingzhou city, Hubei Province, China. After capture, crayfish were immediately transferred to the laboratory. *P. clarkii* were raised separately in six glass tanks (130 cm × 54 cm × 30 cm) for 14 d to acclimate to laboratory conditions before the experiment. During acclimation, the six glass tanks were filled with 20 cm aerated tap water, and PVC pipes and artificial macrophytes (resembling *Hydrilla verticillata*) were used as refuges for crayfish. The initial water temperature was the same as that of the farm, and after 7 d of adaptation, it was then increased by 0.5–1 °C every day to reach the experimental temperature of 24 °C. Crayfish were fed a commercial diet (crude protein content 34% and crude fat content 3%) twice at 8:00 h and 18:00 h every day. The food residuals were removed two hours after feeding. The temperature, pH, and dissolved oxygen were measured twice a day with a YSI ProPlus handheld water quality meter (Thermo Fisher Scientific Company, Waltham, MA, USA), with average values of 24.2 ± 1.0 °C, 7.52 ± 0.21, and 7.51 ± 0.63 mg/L during the acclimation, respectively. The photoperiod followed the natural light and dark cycle. The handling of crayfish in this study followed the ethical regulations of the Institutional Animal Care and Use Committee of the Institute of Hydrobiology, Chinese Academy of Sciences (Approval number: Keshuizhuan 08529).

### 2.2. Oxygen Consumption Rate and Asphyxiation Point Experiment

The oxygen consumption rate and asphyxiation point of juvenile *P. clarkii* and subadult *P. clarkii* (juvenile: 1.17 ± 0.03 g, subadult: 11.68 ± 0.11 g) were measured at five different temperatures (20, 22, 24, 26, and 28 °C) before the hypoxia stress test. For the determination of the oxygen consumption rate, single juvenile crayfish or subadult crayfish were placed into a glass tank of 22 L water (35 cm × 23 cm × 30 cm) sealed with 2 mm liquid paraffin wax, with 30 replicates per temperature level. A dissolved oxygen tester (YSI Pro ODO) was used to determine the variation in dissolved oxygen in water within 24 h. The oxygen consumption rates of juvenile and subadult crayfish were calculated [23,24]. The same method was used for the determination of the asphyxiation point. The dissolved oxygen in the water was determined when the juvenile crayfish and subadult crayfish were in asphyxiation coma (the crayfish lay on its side and did not respond to touching its feet with a glass rod), and the dissolved oxygen in the water was the asphyxiation point of *P. clarkii* at this temperature [23].

### 2.3. Hypoxia Stress Experiment

Three hypoxia stress groups (1, 2, and 3 mg/L) and a normal dissolved oxygen control group (7.5 mg/L) were set for juvenile *P. clarkii* and subadult crayfish, respectively. There were 3 replicas for each treatment group. The whole experiment was carried out in 24 glass tanks (35 cm × 23 cm × 30 cm), and each tank contained 12 crayfish. Before starting the experiment, healthy crayfish (juvenile crayfish: 1.17 ± 0.03 g, subadult crayfish: 11.68 ± 0.11 g) were selected and placed into tanks for 24 h of acclimation. During the acclimation period, the water temperature was 24.3 °C, pH 7.52 ± 0.21, photoperiod L:D = 12 h:12 h. After acclimation, each experimental tank was sealed with 2 mm liquid paraffin. Considering that the content of dissolved oxygen in water would be reduced due to the consumption of experimental animals, nitrogen and oxygen control devices were placed in the tank during sealing to monitor the stability of dissolved oxygen in the water of each experimental group and to ensure that crayfish could not carry out climbing interference experiments [25]. The experiment was carried out for 24 h, and the dissolved oxygen was measured every two hours to adjust the nitrogen to oxygen filling ratio and to maintain the stability of the dissolved oxygen in the water. During the experiment, the actual dissolved oxygen of the three hypoxia stress groups and the control group were 1.05 ± 0.07 mg/L, 2.06 ± 0.10 mg/L, 3.03 ± 0.08 mg/L, and 7.52 ± 0.57 mg/L, respectively.

### 2.4. Sample Collection

Three crayfish were taken from each replicate tankage, for a total of nine shrimp per treatment group, at 3 h, 6 h, 12 h, and 24 h after the beginning of the hypoxia stress experiment, and the hepatopancreases were dissected and preserved in frozen storage tubes. All samples were frozen in liquid nitrogen and then stored at −80 °C to determine the activity of antioxidant enzymes and the expression levels of stress genes.

### 2.5. Enzyme Activity Assay

The hepatopancreas samples were homogenized on ice with 1:9 normal saline volume (sample weight/normal saline volume) and centrifuged at 2500 RPM for 10 min. Supernatant was used for enzyme activity determination. Enzyme-linked immunosorbent assay (ELISA) was used to determine the activities of superoxide dismutase (SOD), catalase (CAT), and lactate dehydrogenase (LDH) and the content of lactic acid (LD) [20,21]. The above operations were carried out in strict accordance with the instructions of the kit. The kit was purchased from Jiancheng Institute of Biological Engineering, Nanjing. The results were read using an enzyme label instrument (SpectraMax@iD3, Molecular Devices, San Jose, CA, USA).

### 2.6. RNA Isolation and Quantitative RT–PCR

We washed the liver and hepatopancreas samples twice with PBS (pH 7.4) and placed the samples in TriPure reagents (Roche, Indianapolis, IN, USA) to extract total RNA. Purity and quantity of isolated RNA were determined by NanoDrop ND-1000 (Thermo Fisher Scientific, Waltham, MA, USA). Random hexamer primers and Moroni mouse leukemia virus reverse transcriptase (Promega, Madison, WI, USA) were used for RNA reverse transcription with OD260/OD280 values between 1.8 and 2.1. Real-time PCR analysis of cDNA samples was conducted in a LightCycler 480 Real Time PCR System (Switzerland, Germany) using FastStart Universal SYBR Green Master Mix (Roche Diagnostics GmbH, Mannheim, Germany). 18S was used as a reference for internal reference genes, and the relative gene expression calculation method was the 2^−ΔΔct^ method [26]. The primer pair sequence is shown in Table 1.

### 2.7. Data Analysis

The oxygen consumption rates, asphyxiation points, and expression levels of antioxidant enzymes, anaerobic metabolic enzymes, and antioxidant genes of *P. clarkii* were expressed as the mean ± standard error (SE). After testing for variance normality and homogeneity, a two-way ANOVA was used to test for differences in oxygen consumption rate and asphyxiation point between juvenile and subadult crayfish at five temperature levels. The activities of antioxidant enzymes (SOD and CAT), the contents of anaerobic metabolizing enzymes (LD and LDH), and the expression levels of the *SOD* mRNA, *CAT* mRNA, *Hsp*70 mRNA, and *crustin 4* mRNA of the four different groups at four different sampling time points were tested for differences using two-way ANOVA. If the difference was significant, an LSD test was used to compare the difference between the two pairs. Differences were considered to be significant if *p* < 0.05. Statistical analysis was performed using SPSS 24.0 (SPSS Inc., Chicago, IL, USA).

## 3. Results

### 3.1. Oxygen Consumption Rate, Asphyxiation Point, and Mortality

The oxygen consumption rate of *P. clarkii* was significantly affected by temperature and size, and there was a significant interaction between them (Table 2). The oxygen consumption rate of juvenile and subadult *P. clarkii* increased with increasing temperatures (20–28 °C). At the same temperature, the oxygen consumption rate of juvenile crayfish was 2.8–2.9 times higher than that of subadult crayfish (all *p* < 0.05) (Figure 1A).

The asphyxiation point was significantly affected by temperature and size, but there was no significant interaction between them (Table 2). The asphyxiation points of juvenile and subadult *P. clarkii* also increased with increasing temperatures (20–28 °C). At the same temperature, the oxygen consumption rate of juvenile crayfish was 2.4–3.0 times higher than that of subadult crayfish (all *p* < 0.05) (Figure 1B).

During the 24 h hypoxia stress experiment, there was no death among the experimental crayfish.

### 3.2. Activities of Antioxidant Enzymes and Anaerobic Metabolic Enzymes in Juvenile P. clarkii

The SOD activity, LD content, and LDH activity of juvenile *P. clarkii* were significantly affected by dissolved oxygen and sampling time, and there was a significant interaction between them (Table 3). The CAT activity of juvenile *P. clarkii* was significantly affected by dissolved oxygen but not by sampling time, and there was a significant interaction between them (Table 3).

There were no significant differences in SOD activity among the three hypoxia groups at 3 h and 24 h (all *p* > 0.05), but all of them were significantly higher than those in the control group (all *p* < 0.05). The activity of SOD in the control group at 6 h and 12 h was significantly lower than that in the 1 mg/L and 2 mg/L groups (all *p* < 0.05), but there was no significant difference between the control group and the 3 mg/L group (all *p* > 0.05). Among the different sampling time points of the same group, there were significant differences in SOD activity in the control group and the 1 mg/L group at the four sampling time points (all *p* < 0.05). The SOD activity of the 2 mg/L and 3 mg/L groups at 3 h and 24 h was significantly higher than that at 6 h and 12 h (all *p* < 0.05), and the SOD activity of the 2 mg/L group at 6 h was significantly higher than that at 12 h (*p* < 0.05) (Figure 2A).

The activity of CAT in the 1 mg/L group was significantly higher than that in the 2 mg/L and 3 mg/L groups and the control group at 3 h, 6 h, and 12 h (*p* < 0.05), and the CAT activity in the 2 mg/L group was significantly higher than that in the control group at 6 h and 24 h (*p* < 0.05). There were no significant differences in CAT activity between the 3 mg/L group and the control group at the four sampling time points (all *p* > 0.05) (Figure 2B).

The LD content of the three hypoxia stress groups was significantly higher than that of the control group at 3 h and 6 h (*p* < 0.05), and the LD content of the 1 mg/L group was significantly higher than that of the 3 mg/L group at 3 h (*p* < 0.05), but there were no significant differences among the four groups at 12 h and 24 h (all *p* > 0.05). Among the different sampling time points of the same group, the LD content of the three hypoxia stress groups at 3 h and 6 h was significantly higher than that at 12 h and 24 h (all *p* < 0.05), while the LD content of the control group had no significant differences at the four sampling time points (all *p* > 0.05) (Figure 2C).

The content of LDH in the 1 mg/L group at 3 h, 6 h, and 12 h was significantly higher than that in the 2 mg/L and 3 mg/L groups and the control group (all *p* < 0.05). The LDH activity in the 2 mg/L and 3 mg/L groups was significantly higher than that in the control group at 3 h (all *p* < 0.05), but there was no significant difference at 6 h and 12 h. Among the different sampling time points of the same group, the LDH activity at 3 h and 6 h in the 1 mg/L group was significantly higher than that at 12 h and 24 h (*p* < 0.05), the LDH activity at 12 h was significantly higher than that at 24 h (*p* < 0.05), and the LDH activity in the 2 mg/L and 3 mg/L groups at 3 h was significantly higher than that at 6 h, 12 h, and 24 h (*p* < 0.05). There were no significant differences in LDH activity in the control group at the four sampling time points (all *p* > 0.05) (Figure 2D).

### 3.3. Activities of Antioxidant Enzymes and Anaerobic Metabolic Enzymes in Subadult P. clarkii 

The activities of SOD, CAT, and LDH and the LD content of subadult *P. clarkii* were significantly affected by dissolved oxygen and sampling time, and there were significant differences (Table 4).

The SOD activity of the 1 mg/L and 2 mg/L groups at 3 h and 6 h was significantly higher than that of the 3 mg/L group and the control group (*p* < 0.05), but there was no significant difference between the latter two groups. The SOD activity of the 1 mg/L and 3 mg/L groups was significantly higher than that of the control group at 12 h, while the SOD activity of the 2 mg/L and 3 mg/L groups was significantly higher than that of the control group at 24 h. Among the different sampling time points of the same group, the SOD activity of the 1 mg/L group at 6 h was significantly higher than that at 3 h, 12 h, and 24 h (*p* < 0.05); the SOD activity of the 2 mg/L group at 24 h was significantly higher than that at 3 h and 12 h; and the SOD activity of the 3 mg/L group at 12 h and 24 h was significantly higher than that at 3 h and 6 h. There were no significant differences in SOD content in the control group at the four sampling time points (all *p* > 0.05) (Figure 3A).

The CAT activity of the 1 mg/L group at 3 h was significantly higher than that of the other three groups (*p* < 0.05), and there were no significant differences among the latter three groups. The CAT activity of the 1 mg/L and 2 mg/L groups at 6 h was significantly higher than that of the 3 mg/L group and control group (*p* < 0.05), while the CAT activity of the three hypoxia groups at 12 h and 24 h was significantly higher than that of the control group (all *p* < 0.05). Among the different sampling time points of the same group, the CAT activity of the 1 mg/L group at 3 h and 6 h was significantly higher than that at 12 h and 24 h (all *p* < 0.05), while the CAT activity of the 2 mg/L group at 24 h was significantly higher than that at 3 h, 6 h, and 12 h (all *p* < 0.05). The CAT activity of the 3 mg/L group increased with the increase in sampling time. There were no significant differences in CAT content in the control group at the four sampling time points (all *p* > 0.05) (Figure 3B).

The LD content of the 1 mg/L and 2 mg/L treatment groups was significantly higher than that of the control group at the four sampling time points (all *p* < 0.05). At 3 h and 6 h, the LD content of the 3 mg/L group was significantly lower than that of the 1 mg/L and 2 mg/L groups, but no significant difference was found between the 3 mg/L group and the control group. The LD content of the 3 mg/L group at 12 h and 24 h was significantly higher than that of the control group (*p* < 0.05). There were significant differences in LD content among the three hypoxia groups at different sampling time points in the same treatment group, and the LD content in the 2 mg/L and 3 mg/L groups increased with time. There were no significant differences in LD content in the control group at the four sampling time points (all *p* > 0.05) (Figure 3C).

The LDH activity of the 1 mg/L and 2 mg/L groups was significantly higher than that of the 3 mg/L group and control group at the four sampling time points (all *p* < 0.05). The LDH activity of the 3 mg/L group was significantly higher than that of control group at 3 h (*p* < 0.05), but there were no significant differences between the 3 mg/L group and control group at 6 h, 12 h, and 24 h. There were significant differences in LD content among the three hypoxia groups at different sampling time points in the same treatment group, and the LD content in the 1 mg/L group decreased with the time of sampling. There were no significant differences in LDH activity in the control group at the four sampling time points (all *p* > 0.05) (Figure 3D).

### 3.4. Antioxidant and Stress Gene Expression of Juvenile P. clarkii 

The expression levels of *SOD* mRNA, *Hsp*70 mRNA, and *crustin 4* mRNA in juvenile *P. clarkii* were significantly affected by dissolved oxygen and sampling time, and there was a significant interaction between them (Table 5). The *CAT* mRNA expression level of juvenile *P. clarkii* was significantly affected by dissolved oxygen but not by sampling time, and there was no significant interaction between them (Table 4).

The *SOD* mRNA expression level of the 1 mg/L and 2 mg/L groups at 3 h was significantly higher than that of the 3 mg/L group and control group (*p* < 0.05), while the *SOD* mRNA expression level of the 1 mg/L group was significantly higher than that of the other three groups at 6 h (*p* < 0.05). There were no significant differences in the *SOD* mRNA expression level among the four groups at 12 h and 24 h. Among the different sampling time points of the same group, the *SOD* mRNA expression level at 6 h in the 1 mg/L group was significantly higher than that at 3 h, 12 h, and 24 h (*p* < 0.05), and the *SOD* mRNA expression level in the 2 mg/L group at 3 h was significantly higher than that at 6 h, 12 h, and 24 h (*p* < 0.05). There were no significant differences in the *SOD* mRNA expression level of the 3 mg/L group and the control group at the four sampling time points (all *p* > 0.05) (Figure 4A).

At the four sampling time points, the *CAT* mRNA expression level of the 1 mg/L group was significantly higher than that of the other three groups (*p* < 0.05), and the 2 mg/L group had no significant differences from the control group. The *CAT* mRNA expression level of the 3 mg/L group was significantly higher than that of the control group only at 24 h (*p* < 0.05). There were no significant differences in the expression of *CAT* mRNA at the four sampling time points for all groups (all *p* > 0.05) (Figure 4B).

The expression of *Hsp*70 mRNA in the 1 mg/L group was significantly higher than that in the other three groups at 3 h and 6 h (*p* < 0.05), and the expression level of *Hsp*70 mRNA in the 1 mg/L and 2 mg/L groups was significantly higher than that in the 3 mg/L group and control group at 12 h (*p* < 0.05). At 24 h, the expression level of *Hsp*70 mRNA in the three hypoxia stress groups was significantly higher than that in the control group (all *p* < 0.05). There were significant differences in the expression of *Hsp*70 mRNA at the four sampling time points for the three hypoxia stress groups (all *p* < 0.05), but there were no significant differences in the expression of *Hsp*70 mRNA in the control group at the four sampling time points (all *p* > 0.05) (Figure 4C).

At the four sampling time points, the expression of *crustin 4* mRNA in the 1 mg/L and 2 mg/L groups was significantly higher than that in the 3 mg/L group and control group (all *p* < 0.05), but there was no significant difference between the 3 mg/L group and control group (*p* < 0.05). There were significant differences in *crustin 4* mRNA expression at different sampling time points for the 1 mg/L and 2 mg/L groups (all *p* < 0.05), but no significant differences were found for the 3 mg/L group and control group at the four sampling time points (all *p* > 0.05) (Figure 4D).

### 3.5. Antioxidant and Stress Gene Expression of Subadult P. clarkii 

The mRNA expression levels of *SOD*, *CAT*, *Hsp70*, and *crustin 4* in subadult *P. clarkii* were significantly affected by dissolved oxygen and sampling time. There was a significant interaction between them for *SOD* mRNA and *CAT* mRNA and no significant interaction between them for *Hsp*70 mRNA and *crustin 4* mRNA (Table 6).

There were no significant differences in the expression of *SOD* mRNA for the four groups at 3 h and 12 h, but the expression of *SOD* mRNA in the 1 mg/L and 2 mg/L groups was significantly higher than that in the 3 mg/L group and control group at 6 h (*p* < 0.05). The *SOD* mRNA expression level in three hypoxia stress groups was significantly higher than that in control group at 24 h (all *p* < 0.05). There were significant differences in *SOD* mRNA expression levels at 6 and 24 h for the three hypoxia stress groups, respectively (all *p* < 0.05), but no significant differences were found in the control group at the four sampling time points (all *p* > 0.05) (Figure 5A).

There were no significant differences in the expression of *CAT* mRNA among the four groups at 3 h, but the expression level of *CAT* mRNA in three hypoxia stress groups was significantly higher than that in the control group at 6 h and 12 h (all *p* < 0.05), and the expression level of *CAT* mRNA in the 1 mg/L and 2 mg/L treatment groups was significantly higher than that in the 3 mg/L group and control group at 24 h (*p* < 0.05). There were significant differences in *CAT* mRNA expression at 6 and 24 h for the three hypoxia stress groups, respectively (all *p* < 0.05). The *CAT* mRNA expression level in the 1 mg/L and 3 mg/L groups increased firstly and then decreased with the increase in sampling time, and the *CAT* mRNA expression level in the 2 mg/L group increased gradually with the increase in sampling time. There were no significant differences in *CAT* mRNA expression levels in the control group at the four sampling time points (all *p* > 0.05) (Figure 5B).

The expression level of *Hsp*70 mRNA in the 1 mg/L group was significantly higher than that in the other three groups at 3 h and 12 h (*p* < 0.05), and there were no significant differences among the latter three groups. The expression level of *Hsp*70 mRNA in three hypoxia stress groups was significantly higher than that in the control group at 6 h and 24 h (all *p* < 0.05), and the expression level of *Hsp*70 mRNA in the 1 mg/L group was significantly higher than that in the 2 mg/L and 3 mg/L groups (*p* < 0.05). There were significant differences in the expression of *Hsp*70 mRNA at the four sampling time points for the three hypoxia stress groups, respectively (all *p* < 0.05), but no significant differences were found in the control group at the four sampling time points (all *p* > 0.05) (Figure 5C).

At the four sampling time points, the expression level of *crustin 4* mRNA in three hypoxia stress groups was significantly higher than that in the control group (all *p* < 0.05), and the expression level of *crustin 4* mRNA in the 1 mg/L group was significantly higher than that in the 3 mg/L group (all *p* < 0.05). There were significant differences in *crustin 4* mRNA expression at the four sampling time points for the three hypoxia stress groups, respectively (all *p* < 0.05), but no significant differences were found in the control group at the four sampling time points (all *p* > 0.05) (Figure 5D).

## 4. Discussion

### 4.1. Effect of Temperature on the Oxygen Consumption Rate and Asphyxiation Point of P. clarkii

Temperature is the most important environmental factor affecting the oxygen consumption rate and asphyxiation point of shrimps and crabs. In the range of 20 °C to 28 °C, the oxygen consumption rate and asphyxiation point of juvenile and subadult *P. clarkii* increased with increasing temperature in the study; the results are consistent with those of *L. vannamei* and *Eriocheir sinensis* (*E. sinensis*) [22,23]. This is because in an appropriate temperature range, the enzyme activity in the tissues of aquatic animals will be enhanced with increasing temperatures, thus accelerating the rate of metabolism and increasing the oxygen consumption and asphyxiation point [27,28]. In addition, individual size will also affect the oxygen consumption level of aquatic animals, and usually, the oxygen consumption rate will decrease with the increasing size of aquatic animals [29]. The results in this experiment are consistent with the above conclusions, which indicated that subadult crayfish had stronger hypoxia tolerance than juvenile crayfish. Under the same conditions, compared with existing large-scale cultured crustaceans, the oxygen consumption rate and asphyxiation point of *P. clarkii* were lower than those of *E. sinensis* [23,30], *Penaeus japonucus* [31], and *L. vannamei* [24]. This shows that *P. clarkii* has a strong ability to adapt to hypoxia.

### 4.2. Effect of Hypoxia on the Activity of Antioxidant and Anaerobic Metabolic Enzymes in P. clarkii

Hypoxia stress can cause an oxidative stress response in *P. clarkii*, disturb the dynamic balance of reactive oxygen species (ROS) in the body, and significantly increase the ROS content in the hepatopancreas, while uncleared ROS will accelerate the formation of intracellular lipid peroxides and cause oxidative damage [32,33]. SOD and CAT are important components of intracellular antioxidant enzymes that scavenge oxygen free radicals. SOD and CAT can convert superoxide free radicals into oxygen molecules and water that are nontoxic to organisms, thus maintaining the normal physiological activities of cells and bodies [34]. In this study, the activities of SOD and CAT in the hypoxia stress groups were higher than those in the control group for both juvenile crayfish and subadult crayfish, indicating that hypoxia stress could induce an increase in SOD and CAT activities in the hepatopancreas of both groups to address the oxidative damage caused by ROS. The results are consistent with the results of *L. vannamei* [20]. Under the same conditions, the oxygen consumption rate of juvenile shrimps was significantly higher than that of subadult shrimps, resulting in significantly increased SOD and CAT enzyme activities in the hepatopancreas of juvenile shrimps and advanced oxidative stress reactions [20,29].

Previous studies have shown that when *P. clarkii* is exposed to hypoxia, the pathway of energy metabolism changes from aerobic respiration to anaerobic respiration to maintain physiological balance [35,36]. The main end product of anaerobic respiration in crustaceans is lactic acid (LD), which can enhance the binding ability of hemocyanin and oxygen molecules in hypoxia [37,38,39], while LDH is one of the important enzymes in anaerobic glycolysis and gluconeogenesis, which can catalyze the redox reaction between LD and pyruvate [40]. In this study, the LD content and LDH activity in the hepatopancreas of juvenile crayfish showed significant fluctuations in a short period of time, while the fluctuations in LD content and LDH activity in the hepatopancreas of subadult crayfish were weaker than those of juvenile crayfish. The reason for this difference is that under the same conditions, the brain, hepatopancreas, and other organs and tissues that maintain life activities in juvenile shrimp account for a larger proportion of body mass [41]. These organs consume more energy for metabolic activities, increase anaerobic respiration, and result in higher LD content and LDH activity in the hepatopancreas of juvenile shrimp compared to subadult shrimp.

Under hypoxia stress, juvenile crayfish will rapidly increase the anaerobic respiration rate in a short time (6 h) to generate energy and maintain body balance, which will lead to a rapid increase in LD content and a significant increase in LDH activity, while excessive LDH activity will cause liver damage, and the higher the activity of LDH, the greater the degree of hepatopancreas injury [39]. When the activity of LDH is too high, the degree of liver injury increases. To protect the liver from injury, the body inhibits anaerobic respiration, thus reducing the LD content and LDH activity to protect the body. In summary, juvenile crayfish are more susceptible to hypoxic conditions than subadult crayfish at the same temperature, and anaerobic respiration causes more serious damage to juvenile crayfish.

### 4.3. Effect of Hypoxia on the Expression of Antioxidant and Stress Genes in P. clarkii

When the metabolic balance of ROS in aquatic animals is disrupted, oxidative stress reactions are induced. Excessive accumulation of ROS leads to damage to cell proteins and lipid membranes, while SOD and CAT are the first line of defense against oxidative stress in biological organisms [42]. Compared with the control group, the expression of *SOD* mRNA and *CAT* mRNA in the hypoxia stress groups was significantly increased. The expression of *SOD* mRNA and *CAT* mRNA in the 1 mg/L group was higher than that in the 2 mg/L and 3 mg/L groups at 3 h and 6 h, and the expression levels of *SOD* mRNA and *CAT* mRNA in juvenile crayfish in the 1 mg/L group were higher than those in subadult crayfish. The results showed that the stronger the degree of hypoxia stress, the higher the expression levels of *SOD* mRNA and *CAT* mRNA of *P. clarkii* in a short time (6 h). Under the same hypoxia stress, the oxidative stress response of juvenile crayfish was stronger than that of subadult crayfish.

The expression level of *Hsp*70 mRNA can reflect the effects of environmental stress on the body. *Hsp*70 mRNA can play an important biological function under stress conditions and is an activator of innate immunity, playing a key role in various stress responses and alleviating stress-induced injury [43,44,45]. In this study, the *Hsp*70 mRNA expression levels of juvenile and subadult crayfish in the hypoxia stress groups were significantly increased compared to those in the control group. The *Hsp*70 mRNA expression levels in juvenile and subadult crayfish in the 1 mg/L group were higher than those in the 2 mg/L and 3 mg/L groups, and the *Hsp*70 mRNA expression level of juvenile crayfish was higher than that of subadult crayfish under the same hypoxia stress. The results showed that the expression level of *Hsp*70 mRNA in juvenile and subadult crayfish was positively correlated with the degree of hypoxia stress, with the latter showing lower stress levels under the same hypoxia stress conditions.

Crustin 4 is an important immune effector molecule that is an important component of pathogen resistance and plays a crucial role in crayfish resistance to bacterial infection [46,47]. In this study, the expression level of *crustin 4* mRNA in the 1 mg/L group was higher than that in the 2 mg/L and 3 mg/L groups, and the *crustin 4* mRNA expression level of subadult crayfish was generally higher than that of juvenile crayfish, indicating that the stronger the degree of hypoxia stress level, the higher the expression level of *crustin 4* mRNA. In the same hypoxic environment, juvenile crayfish have weaker resistance to bacterial infection than subadult crayfish and are more susceptible to disease.

For a short period of time (24 h), whether juvenile crayfish or subadult crayfish are exposed to a low dissolved oxygen environment, *P. clarkii* will always be in a state of stress, thus inducing the organism to increase the activities of SOD and CAT antioxidant enzymes and gene expression levels and improve the immune capacity of the organism to combat the oxidative damage caused by stress and infection by pathogenic bacteria. However, the enhancement of antioxidant capacity and immune capacity leads to an increase in energy consumption. When the energy generated by aerobic and anaerobic respiration is not sufficient to maintain a high level of immunity, the probability of pathogens from the environment invading the organism increases, resulting in an increase in the incidence or probability of death in crayfish [47]. In addition, when the organism is in a hypoxic environment for a long time, the accumulated oxidation products in the body beyond the tolerance range of the organism will lead to damage to the immune and antioxidant system and eventually cause disease and economic losses [32].

## 5. Conclusions

Red swamp crayfish *P. clarkii* had a low oxygen consumption rate and asphyxiation point and no death in the 24 h 1 mg/L hypoxia stress experiment, which indicated that *P. clarkii* had strong hypoxia adaptability, and the hypoxia adaptability of subadult crayfish was stronger than that of larvae. Hypoxia stress enhanced the antioxidant responses of crayfish and induced the upregulation of antioxidant genes to adapt to hypoxia. Compared to subadult crayfish, the antioxidant system of juvenile crayfish is more sensitive to hypoxic environments and is more likely to produce an oxidative stress response, resulting in increased probability of pathogen infection. The present results are from a 24 h acute hypoxia stress experiment; however, further studies on the effects of intermittent hypoxia stress on the survival, growth, and physiological function of *P. clarkii* on a longer time scale are needed in the future.

## Figures and Tables

**Figure 1 biology-13-00033-f001:**
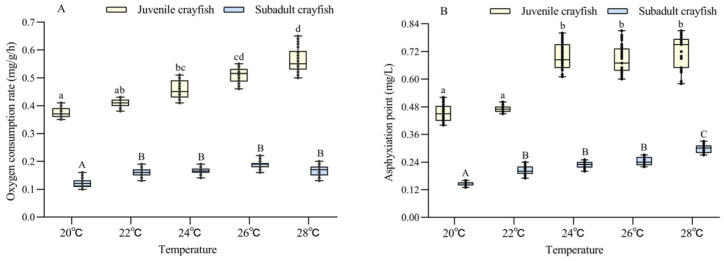
Analysis of the difference in oxygen consumption rate (**A**) and asphyxiation point (**B**) between juvenile and subadult *Procambarus clarkii* at different temperature levels. There were 30 shrimp per temperature level, and each point showed the individual’s oxygen consumption rate and asphyxiation point. In the figure, there were no significant differences in oxygen consumption rate and asphyxiation point between the treatment groups with the same letters, but there were significant differences between the treatment groups with different letters (*p* < 0.05).

**Figure 2 biology-13-00033-f002:**
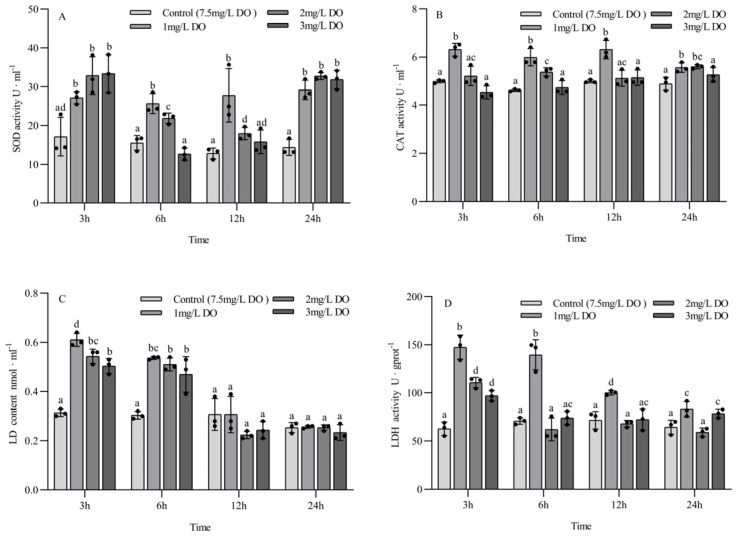
Changes in antioxidant enzyme and respiratory enzyme activities of juvenile *Procambarus clarkii* under different dissolved oxygen levels and different sampling times. Superoxide dismutase (**A**), Catalase (**B**), Lactate content (**C**), Lactate dehydrogenase (**D**). There were three repeat groups for each dissolved oxygen level, replicate groups of 12 shrimp, with water temperatures of 24.0 °C, and each point showed the average enzyme activity of the three shrimp under the current conditions. In the figure, there were no significant differences in enzyme activity between the groups with the same letters, but there were significant differences between the groups with different letters (*p* < 0.05).

**Figure 3 biology-13-00033-f003:**
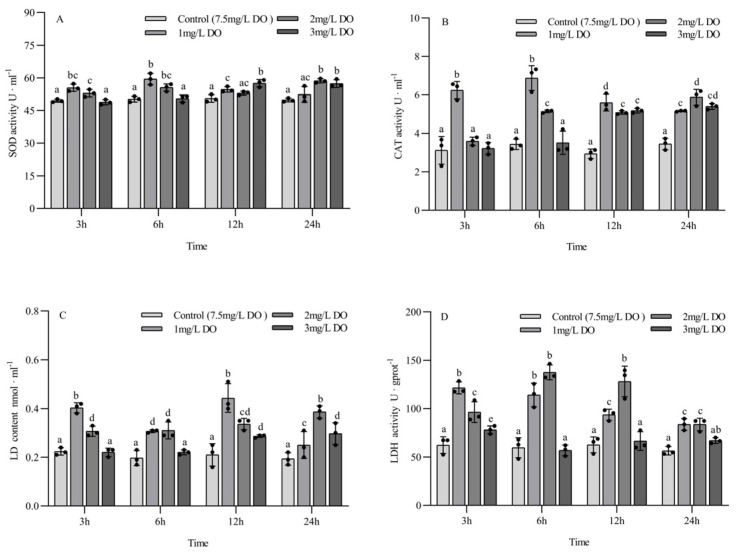
Changes in antioxidant enzyme and respiratory enzyme activities of subadult *Procambarus clarkii* under different dissolved oxygen levels and different sampling times. Superoxide dismutase (**A**), Catalase (**B**), Lactate content (**C**), Lactate dehydrogenase (**D**). There were three repeat groups for each dissolved oxygen level, replicate groups of 12 shrimp, with water temperatures of 24.0 °C, and each point showed the average enzyme activity of the three shrimp under the current conditions. In the figure, there were no significant differences in enzyme activity between the groups with the same letters, but there were significant differences between the groups with different letters (*p* < 0.05).

**Figure 4 biology-13-00033-f004:**
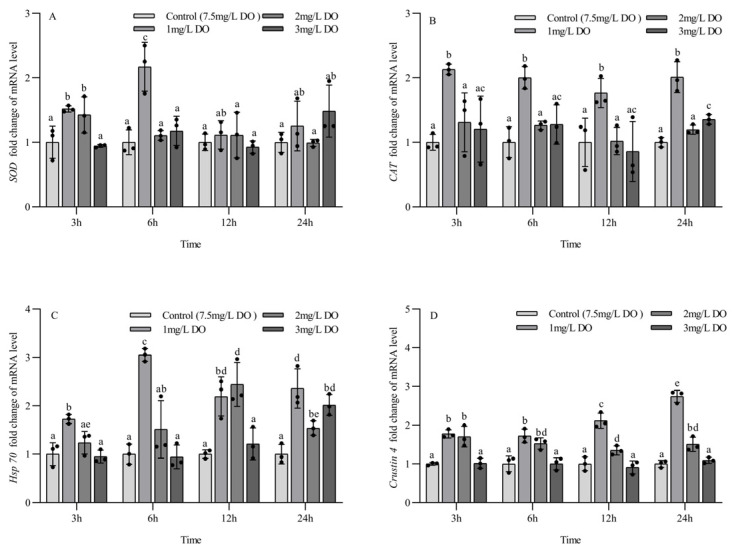
Changes in antioxidant and stress genes in juvenile *Procambarus clarkii* under different dissolved oxygen levels and different sampling times. Superoxide dismutase mRNA (**A**), Catalase mRNA (**B**), Heat shock protein 70 mRNA (**C**), Crustin 4 mRNA (**D**). There were three repeat groups for each dissolved oxygen level, replicate groups of 12 shrimp, with water temperatures of 24.0 °C, and each point showed the average of the gene expression levels of the three shrimp under the current conditions. In the figure, there were no significant differences in gene expression between the treatment groups with the same letters, but there were significant differences between the treatment groups with different letters (*p* < 0.05).

**Figure 5 biology-13-00033-f005:**
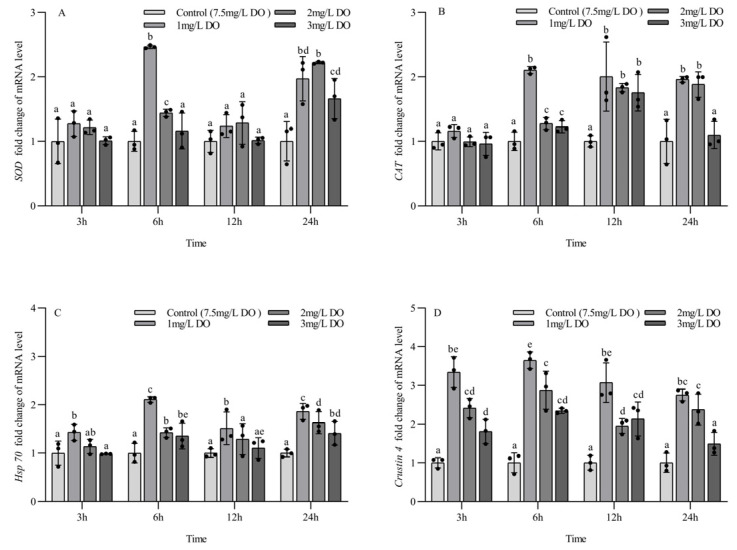
Changes in antioxidant and stress genes in subadult *Procambarus clarkii* under different dissolved oxygen levels and different sampling times. Superoxide dismutase mRNA (**A**), Catalase mRNA (**B**), Heat shock protein 70 mRNA (**C**), Crustin 4 mRNA (**D**). There were three repeat groups for each dissolved oxygen level, replicate groups of 12 shrimp, with water temperatures of 24.0 °C, and each point showed the average of the gene expression levels of the three shrimp under the current conditions. In the figure, there were no significant differences in gene expression between the treatment groups with the same letters, but there were significant differences between the treatment groups with different letters (*p* < 0.05).

**Table 1 biology-13-00033-t001:** Reference sources for qPCR primers.

Gene	Sequence(5′-3′)	GenBank Accession No.
18S-F	GTCAGGTCATCACCATCGGCA	HQ-414542.1
18S-R	CGGTCTCGTGAACACCAGCA	
SOD-F	CGCCGATGTAAGACTGGGACG	TRINITY-DN63424-c1-g1
SOD-R	CTCCAGGTAAACACGGCTTCCAT	
CAT-F	TCCTGTGAACTGTCCCTATCGTG	TRINITY-DN27834-c0-g1
CAT-R	AACCCAGTCTTCTTACAATCAACG	
Crustin-F	CTCTGACTGCCAGGTGTTT	NW-020872843.1
Crustin-R	TGCGAGCTGTGATGGTTAG	
*Hsp*70-F	GTTGACCAAGATGAAGGAGAC	DQ-301506.1
*Hsp*70-R	CTGACGCTGAGAGTCGTTG	

**Table 2 biology-13-00033-t002:** Two-way ANOVA for oxygen consumption rate and asphyxiation point of juvenile and subadult *Procambarus clarkii* at five temperature (20, 22, 24, 26, 28 °C) levels.

Variable of Treatment	F Value	df	*p* Value
	Oxygen consumption rate		
Temperature	13.26	4	<0.001
Size	656.42	1	<0.001
Temperature × Size	5.17	4	0.005
	Asphyxiation point		
Temperature	11.93	4	<0.001
Size	260.70	1	<0.001
Temperature × Size	2.71	4	0.059

**Table 3 biology-13-00033-t003:** Two-way ANOVA for the SOD, CAT, LD, and LDH of juvenile *Procambarus clarkii* among four dissolved oxygen treatments (1, 2, 3, 7.5 mg/L) at four sampling times (3, 6, 12, 24 h).

Variable of Treatment	F Value	df	*p* Value
	SOD		
Dissolved oxygen	36.62	3	<0.001
Sampling time	28.13	3	<0.001
Dissolved oxygen × Sampling time	7.61	9	<0.001
	CAT		
Dissolved oxygen	43.53	3	<0.001
Sampling time	1.85	3	0.157
Dissolved oxygen × Sampling time	6.54	9	<0.001
	LD		
Dissolved oxygen	27.41	3	<0.001
Sampling time	139.05	3	<0.001
Dissolved oxygen × Sampling time	12.52	9	<0.001
	LDH		
Dissolved oxygen	88.67	3	<0.001
Sampling time	36.69	3	<0.001
Dissolved oxygen × Sampling time	12.78	9	<0.001

**Table 4 biology-13-00033-t004:** Two-way ANOVA for the SOD, CAT, LD, and LDH of subadult *Procambarus clarkii* among four dissolved oxygen treatments (1, 2, 3, 7.5 mg/L) at four sampling times (3, 6, 12, 24 h).

Variable of Treatment	F Value	df	*p* Value
	SOD		
Dissolved oxygen	24.08	3	<0.001
Sampling time	6.30	3	0.002
Dissolved oxygen × Sampling time	10.19	9	<0.001
	CAT		
Dissolved oxygen	109.87	3	<0.001
Sampling time	13.51	3	<0.001
Dissolved oxygen × Sampling time	15.15	9	<0.001
	LD		
Dissolved oxygen	55.17	3	<0.001
Sampling time	7.46	3	0.001
Dissolved oxygen × Sampling time	8.49	9	<0.001
	LDH		
Dissolved oxygen	109.85	3	<0.001
Sampling time	12.91	3	<0.001
Dissolved oxygen × Sampling time	10.41	9	<0.001

**Table 5 biology-13-00033-t005:** Two-way ANOVA for the *SOD* mRNA, *CAT* mRNA, *Hsp*70 mRNA, and *crustin 4* mRNA of juvenile *Procambarus clarkii* among four dissolved oxygen treatments (1, 2, 3, 7.5 mg/L) at four sampling times (3, 6, 12, 24 h).

Variable of Treatment	F Value	df	*p* Value
	*SOD* mRNA		
Dissolved oxygen	10.08	3	<0.001
Sampling time	3.75	3	0.020
Dissolved oxygen × Sampling time	4.35	9	0.001
	*CAT* mRNA		
Dissolved oxygen	30.10	3	<0.001
Sampling time	2.63	3	0.066
Dissolved oxygen × Sampling time	0.47	9	0.883
	*Hsp*70 mRNA		
Dissolved oxygen	46.64	3	<0.001
Sampling time	7.70	3	0.001
Dissolved oxygen × Sampling time	7.13	9	<0.001
	*Crustin 4* mRNA		
Dissolved oxygen	130.75	3	<0.001
Sampling time	8.33	3	<0.001
Dissolved oxygen × Sampling time	8.17	9	<0.001

**Table 6 biology-13-00033-t006:** Two-way ANOVA for the *SOD* mRNA, *CAT* mRNA, *Hsp*70 mRNA, and *crustin 4* mRNA of subadult *Procambarus clarkii* among four dissolved oxygen treatments (1, 2, 3, 7.5 mg/L) at four sampling times (3, 6, 12, 24 h).

Variable of Treatment	F Value	df	*p* Value
	*SOD* Mrna		
Dissolved oxygen	27.30	3	<0.001
Sampling time	21.46	3	<0.001
Dissolved oxygen × Sampling time	6.70	9	<0.001
	*CAT* Mrna		
Dissolved oxygen	33.05	3	<0.001
Sampling time	19.24	3	<0.001
Dissolved oxygen × Sampling time	5.20	9	<0.001
	*Hsp*70 Mrna		
Dissolved oxygen	27.74	3	<0.001
Sampling time	8.73	3	<0.001
Dissolved oxygen × Sampling time	1.61	9	0.154
	*Crustin 4* Mrna		
Dissolved oxygen	78.54	3	<0.001
Sampling time	5.32	3	0.004
Dissolved oxygen × Sampling time	1.53	9	0.178

## Data Availability

The data that support the findings of this study are available from the corresponding author upon reasonable request.

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
