# Peer review of "Effects of Hypoxia Stress on Survival, Antioxidant and Anaerobic Metabolic Enzymes, and Related Gene Expression of Red Swamp Crayfish *Procambarus clarkii"

_biology, 2024, doi:10.3390/biology13010033_

Round 1
Reviewer 1 Report
Comments and Suggestions for Authors
In my opinion, MS is of high quality and I think it should be accepted after minor revision. In the introduction (at the end, next to the MS objectives), I suggest adding research hypotheses. In the Material and Methods section I propose additions:
2.1. Add information whether the breeding stock of crayfish was bred or obtained from the wild. Provide the commercial feed protein and fat levels. If you measured ammonia and nitrite - please add information.
2.2. How the crayfish acclimatized to different temperatures. Please describe or provide reference(s).
2.6. The table title should be more informative. It should be remembered that each table or figure can have a so-called "own" life.
In the Results, I only suggest that Latin names (in tables and figures) should be written as full names, not abbreviated ones.
I also suggest changing the words "we", "our" into the passive voice, e.g. it was; present results, ect ......
For all used equipment, especially measurements, check that the manufacturer, city and country are listed.
Author Response
|
Comments 1: In the introduction (at the end, next to the MS objectives), I suggest adding research hypotheses. |
|
Response 1: Thank you for your suggestions. We have supplemented the research hypothesis in the introduction. “Thus, it was hypothesized that the activities of antioxidant enzymes, anaerobic respiratory enzymes and heat stress gene expression levels of P.clarkii increased under anoxic environment, and there were differences between juvenile and subadult P.clarkii.” Please see the lines 102-104 in the revised document. |
|
Comments 2: Add information whether the breeding stock of crayfish was bred or obtained from the wild. Provide the commercial feed protein and fat levels. If you measured ammonia and nitrite - please add information. |
|
Response 2: Thank you for your suggestions. We have supplemented the information of experimental crayfish and the protein and fat levels of the commercial feed in the section of materials and methods. “Cultured juvenile crayfish (1.05 ±0.05 g) and subadult crayfish (11.21 ± 0.15 g) were obtained from the crayfish breeding farm of Yangtze University in Jingzhou city, Hubei Province, China.” “Crayfish were fed a commercial diet (crude protein content 34% and crude fat content 3%) twice at 8:00 h and 18:00 h every day. Please see the lines 120-122 and lines 128-130 in the revised document, respectively. |
|
Comments 3: How the crayfish acclimatized to different temperatures. Please describe or provide reference(s). |
|
Response 3: Thank you for your suggestions. We have supplemented the specific methods for the experimental crayfish to adapt to the temperature during acclimation. “The initial water temperature is the same as that of the farm, and after 1 week of adaptation, it is then increased by 0.5-1 ℃ every day to reach the experimental temperature of 24 ℃.” Please see the lines 126-128 in the revised document. |
|
Comments 4: The table title should be more informative. It should be remembered that each table or figure can have a so-called "own" life. |
|
Response 4: Thank you for your suggestions. We totally agree. We have revised the titles of all table and figures based on your suggestion in the revised document. Please see the tables and figures in the revised document. |
|
Comments 5: In the Results, I only suggest that Latin names (in tables and figures) should be written as full names, not abbreviated ones. |
|
Response 5: Thank you for your suggestions. We totally agree. We have revised the titles of all table and figures based on your suggestion in the revised document. Please see the tables and figures in the revised document. |
|
Comments 6: I also suggest changing the words "we", "our" into the passive voice, e.g. it was; present results, ect ...... |
|
Response 6: Thank you for your suggestions. We have revised based on your suggestion. “Thus, it was hypothesized that the activities of antioxidant enzymes, anaerobic respiratory enzymes and heat stress gene expression levels of P.clarkii increased under anoxic environment,” “Present results was a 24-hour acute hypoxia stress experiment;” Please see the lines 102-104 and lines 631 in the revised document, respectively. |
|
Comments 7: For all used equipment, especially measurements, check that the manufacturer, city and country are listed. |
|
Response 7: Thank you for your suggestions. We have supplemented the information of experimental equipment in the section of materials and methods. “The temperature, pH and dissolved oxygen were measured twice a day with a YSI ProPlus handheld water quality meter (Thermo Fisher Scientific Company, Waltham, USA), ” “The results were read using an enzyme label instrument (SpectraMax@iD3, Molecular Devices, SV, USA).” Please see the lines 130-132 and lines 217-218 in the revised document, respectively. |
Reviewer 2 Report
Comments and Suggestions for Authors
The manuscript was well revised by authors. However, some miscellaneous points should be addressed before consideration for publication.
1. Please remove keywords that are already included in the title. You may add further keywords to replace the ones removed if you wish.
2. The first time you use an abbreviation, it's important to spell out the full term and put the abbreviation in parentheses.
3. Provide common name and scientific name of the species you mentioned in parentheses.
4. How to prepare the sandfish before sampling?
5. Replace p-value “0.000” by “<0.001”
6. The reference item 44 was not cited in text.
Comments on the Quality of English Language
Minor editing of English language required.
Author Response
|
Comments 1: Please remove keywords that are already included in the title. You may add further keywords to replace the ones removed if you wish. |
|
Response 1: Thank you for pointing that out. We agree with this comment. Therefore, we modified the keywords. “Environmental stress; oxygen consumption rate; enzyme activity; Stress genes; Crayfish” Please see page 1, line 33 of the article for details. |
|
Comments 2: The first time you use an abbreviation, it's important to spell out the full term and put the abbreviation in parentheses. |
|
Response 2: “Thank you for pointing that out. We agree with this comment. Therefore, we modified the abbreviations of the terms according to the specification. For details,” Please see line 25-26, on page 1, line 52, line 77, line 93 on page 2, line 522 on page 13 of the article. |
|
Comments 3: Provide common name and scientific name of the species you mentioned in parentheses. |
|
Response 3: Thank you for pointing that out. Species names in parentheses in the full text are Crayfish (Procambarus clarkii), Virile crayfish (Orconectes virilis), White shrimp (Litopenaeus vannamei), Oriental river prawn (Macrobrachium nipponense), Chinese ritten crab (Eriocheir sinensis). |
|
Comments 4: How to prepare the sandfish before sampling? |
|
Response 4: Thank you for your question, after sample selection, shrimp were placed into a water body containing anesthetic MS-222 for post-anesthetic sampling. |
|
Comments 5: Replace p-value “0.000” by “<0.001” |
|
Response 5: Thank you for pointing this out. We have made changes to the data in the full table, Please see the tables and figures in the revised document. as shown in the article on lines 294-295 on page 6, 345-346 on page 7, 403-404 on page 9, 457-458 on page 11, 515-516 on page 13, |
|
Comments 6: The reference item 44 was not cited in text. |
|
Response 6: Thank you for pointing this out. We have checked all the literature, and reference 44 has been revised. Please see page 14 line 593. |
|
4. Response to Comments on the Quality of English Language |
|
Point 1: Minor editing of English language required |
|
Response 1: Thank you for your suggestion, the language has been checked and modified. |
Reviewer 3 Report
Comments and Suggestions for Authors
This study is aimed at assessing the physiological response to different hypoxia conditions (1 mg/L; 2 mg/L and 3 mg/L of dissolved oxygen) of juvenile and subadult crayfish reared at different temperatures. Hypoxia is a stressing conditions that can occur in aquaculture, causing disease and death. Hypoxia caused increases in SOD, CAT, LDH activities as well as in the content of LD both in juveniles and subadults. The expression of related genes was also increased, but no mortality was recorded, suggesting that red swamp crayfish is able to cope with hypoxia stress. This study is interesting, the results and discussion are consistent with the objectives and I think that this article can be considered for publication in Biology, after minor revisions as suggested in the specific comments below.
Abstract, line 15, is the most reared (not productive) shrimp species in China;
line 27, the superoxide dismutase (SOD), catalase (CAT), lactate dehydrogenase (LDH) and the content of lactic acid (LD) must be reported in full at their first indication, and further abbreviated.
Introduction, pag 2, line 48, the main modality of P. clarkii culture; is integrated rice-crayfish; line 50, rice fields, so affecting the......and restricting....;
line 58, the deterioration of the quality of the culture environment
line 59, and throughout the text, ammonia stress (without nitrogen),
line 65 and line 68, high nitrite (without nitrogen)
line 74, adaptability to hypoxia
line 77, crustacean mainly show innate immune response
line 80, remove superoxide dismutase
line 86, remove cyclic hypoxia (not necessary to repeat)
line 88, instead In the hypoxic environment I suggest Under hypoxic conditions, the intestinal...
line 92, decreased, and the normal metabolic activity changed
line 95, under hypoxia
line 99, enzymes under hypoxic conditions
line 105, crayfish under the same hypoxic stress......The results of this investigation help to reveal
line 107, survival rate of P. clarkii in farming.
Materials and methods
line 118, The food residuals were removed two hours after feeding
line 122, handling of crayfish in this study
pag. 3, line 4, during sealing to monitor the stability of dissolved oxygen
paragraph. 2.4, were taken from each replicate tahnk
paragraph 2.5, normal saline volume, and centrifuged at 2500 RPM.
The results were read using an enzyme label instrument
caption to figures 1 and others: there were 30 shrimps; temperature level;what do you mean with choking point? please explain better or remove
paragraph 3.2, line 11, the sentence (Among the different sampling time points of the same group.....) is in contrast with the different letters indicated in figure 2A, please check or remove
paragraph 3.3 The activities.....were significantly affected
paragraph 3.4, line 10, four groups at 12 h (not at 24 h too, please check in figure)
paragraph 3.5, line 11, SOD mRNA expression levels at 6 and 24 h (not at four sampling times); also at line 19, CAT mRNA expression at 6 and 24 h (not at the four sampking times)
I suggest to uniform the scale of y-axis of Figures 4 and 5 (graph C), to make comparison of juveniles and subadults easier
pag. 13, last line, juvenile crayfish were smaller (I suppose is subadult crayfish, please check)
pag. 14, line 4, in the hepatopnacreas of juvenile shrimp compared to subadult shrimp (please check). Under hypoxia stress (instead of In the face of....
Comments on the Quality of English Language
some minor points are reported in the comments to the authors
Author Response
|
Comments 1: Abstract, line 15, is the most reared (not productive) shrimp species in China; |
|
Response 1: Thank you for pointing this out. We have made a change. “The red swamp crayfish Procambarus clarkii is the most reared shrimp in China,” Please see page 1, line 13. |
|
Comments 2: line 27, the superoxide dismutase (SOD), catalase (CAT), lactate dehydrogenase (LDH) and the content of lactic acid (LD) must be reported in full at their first indication, and further abbreviated. |
|
Response 2: Thank you for pointing this out. We have made changes. “The activities of the superoxide dismutase (SOD), catalase (CAT), lactate dehydrogenase (LDH) and the content of lactic acid (LD) in the hepatopancreas of juvenile and subadult crayfish in the hypoxia stress groups increased significantly.” Please see lines 25-27 on page 1. |
|
Comments 3: Introduction, pag 2, line 48, the main modality of P. clarkii culture; is integrated rice-crayfish; line 50, rice fields, so affecting the......and restricting....; |
|
Response 3: Thank you for pointing this out. We have made changes. “the main mode of Procambarus clarkii (P.clarkii) culture in China is integrated rich-crayfish.” Please see lines 52 on page 2. |
|
Comments 4: line 58, the deterioration of the quality of the culture environment |
|
Response 4: Thank you for pointing this out. We have made changes. “Previous studies have shown that the deterioration of the quality of the aquaculture environment poses a great challenge to the survival of crustaceans,” Please see lines 61-62 on page 2. |
|
Comments 5: line 59, and throughout the text, ammonia stress (without nitrogen), |
|
Response 5: Thank you for pointing this out. We have made changes. “including temperature stress [7], pH stress [8], ammonia stress [9], hypoxia stress [10] and other factors that often lead to disease and death of crustaceans.” Please see lines 63 on page 2. |
|
Comments 6: line 65 and line 68, high nitrite (without nitrogen) |
|
Response 6: Thank you for pointing this out. We have made changes. “Therefore, the phenomena of low pH, high nitrite and low dissolved oxygen often oc-cur during the culture of P.clarkii in rice fields” Please see lines 69 on page 2. |
|
Comments 7: line 74, adaptability to hypoxia |
|
Response 7: Thank you for pointing this out. We have made changes. “there are few studies on the adaptability to hypoxia of P.clarkii and the effects of hypoxia stress on the immune antioxidant system and related gene expression.” Please see lines 78 on page 2. |
|
Comments 8: line 77, crustacean mainly show innate immune response |
|
Response 8: Thank you for pointing this out. We have made changes. “Crustaceans mainly show innate immune response to hypoxia stress, including physical defense and humoral immunity” Please see lines 82 on page 2. |
|
Comments 9: line 80, remove superoxide dismutase |
|
Response 9: Thank you for pointing this out. We have made changes. “Previous studies have shown that hypoxia stress can increase the content of superoxide anion in the hepatopancreas and the activity of SOD in the gills and hepatopancreas of L.vannamei” Please see lines 84 on page 2. |
|
Comments 10: line 86, remove cyclic hypoxia (not necessary to repeat) |
|
Response 10: Thank you for pointing this out. We have made changes. “Short-term (≤7 days) hypoxia stress can increase the expression of HIF-1a and enhance the antioxidant response in the hepatopancreas and midgut of L.vannamei,” Please see lines 87 on page 2. |
|
Comments 11: line 88, instead In the hypoxic environment I suggest Under hypoxic conditions, the intestinal... |
|
Response 11: Thank you for pointing this out. We have made changes. “Under hypoxic conditions, the intestinal flora and mucosal morphology of Macrobrachium nipponense (M.nipponense) changed” Please see lines 92 on page 2. |
|
Comments 12: line 92, decreased, and the normal metabolic activity changed |
|
Response 12: Thank you for pointing this out. We have made changes. “The enzyme activity related to glucose anaerobic glycolysis increased, the mRNA ex-pression level of aerobic respiratory decreased, and the normal metabolic activity changed from aerobic respiration to anaerobic respiration,” Please see lines 94-96 on page 2. |
|
Comments 13: line 95, under hypoxia |
|
Response 13: Thank you for pointing this out. We have made changes. “which disturbed the energy metabolism of M.nipponense and affected the defense regulation of the antioxidant system under hypoxia environment” Please see lines 98 on page 2. |
|
Comments 14: line 99, enzymes under hypoxic conditions |
|
Response 14: Thank you for pointing this out. We have made changes. “In this study, to explore the hypoxia adaptability of P.clarkii and to understand the changes in the antioxidant system and anaerobic metabolic enzymes under hypoxic conditions,” Please see lines 104-108 on page 2-3. |
|
Comments 15: line 105, crayfish under the same hypoxic stress......The results of this investigation help to reveal |
|
Response 15: Thank you for pointing this out. We have made changes. “The differences in the antioxidant systems and anaerobic metabolic enzymes between juvenile crayfish and subadult crayfish under the same hypoxic stress were compared. The results of this investigation help to reveal the physiological and biochemical reactions of P.clarkii under hypoxic conditions,” Please see lines 112-115 on page 3. |
|
Comments 16: line 107, survival rate of P. clarkii in farming. |
|
Response 16: Thank you for pointing this out. We have made changes. “providing a theoretical basis for improving the survival rate of P.clarkii in farming.” Please see lines 116-117 on page 3. |
|
Comments 17: line 118, The food residuals were removed two hours after feeding |
|
Response 17: Thank you for pointing this out. We have made changes. “Crayfish were fed a commercial diet (crude protein content 34% and crude fat content 3%) twice at 8:00 h and 18:00 h every day. The food residuals were removed two hours after feeding.” Please see lines 128-130 on page 3. |
|
Comments 18: line 122, handling of crayfish in this study |
|
Response 18: Thank you. We have made changes. “The handling of crayfish in this study followed the ethical regulations of the Institutional Animal Care and Use Committee of the Institute of Hydrobiology, Chinese Academy of Sciences” Please see lines 134-136 on page 3. |
|
Comments 19: pag. 3, line 4, during sealing to monitor the stability of dissolved oxygen |
|
Response 19: Thank you. We have made changes. “Considering that the content of dissolved oxygen in water would be reduced due to the consumption of experimental animals, nitrogen and oxygen control devices were placed in the tank during sealing to monitor the stability of dissolved oxygen in the water of each experimental group and ensure that crayfish could not carry out climbing interference experiments” Please see lines 194-198 on page 4. |
|
Comments 20: paragraph. 2.4, were taken from each replicate tahnk |
|
Response 20: Thank you. We have made changes. “Three crayfish were taken from each replicate tankage,” Please see lines 204 on page 4. |
|
Comments 21: paragraph 2.5, normal saline volume, and centrifuged at 2500 RPM. |
|
Response 21: Thank you for pointing this out. We have made changes. “The hepatopancreas samples were homogenized on ice with 1 : 9 normal saline volume (sample weight: normal saline volume), and centrifuged at 2500 RPM for 10 minutes.” Please see lines 210-212 on page 4. |
|
Comments 22: The results were read using an enzyme label instrument |
|
Response 22: Thank you. We have made changes. “The results were read using an enzyme label instrument (SpectraMax@iD3, Molecular Devices, SV, USA).” Please see lines 217-218 on page 4. |
|
Comments 23: caption to figures 1 and others: there were 30 shrimps; temperature level; what do you mean with choking point? please explain better or remove |
|
Response 23: Thank you for pointing this out. We have made changes. “There were 30 shrimp per temperature lever, and each point showed the individual's oxygen consumption rate and asphyxiation point. In the figure, there were no significant differences in oxygen consumption rate and asphyxiation point between the treatment groups with the same letters, but there were significant differences between the treatment groups with different letters (P < 0.05).” Please see lines 285-289 on page 5. Water temperature and shrimp quantity were added in the figure title, see the modified document for details. |
|
Comments 24: paragraph 3.2, line 11, the sentence (Among the different sampling time points of the same group.....) is in contrast with the different letters indicated in figure 2A, please check or remove |
|
Response 24: Thank you for pointing this out. We have made changes. “Among the different sampling time points of the same group, there was significant difference in SOD activity in the control group and 1 mg/L group at four sampling time points (all P < 0.05).” Please see lines 306-308 on page 6. |
|
Comments 25: paragraph 3.3 The activities.....were significantly affected |
|
Response 25: Thank you. We have made changes. “The activities of SOD, CAT and LDH and LD content of subadult P.clarkii was significantly affected by dissolved oxygen and sampling time, and there was a significant affected” Please see lines 349-351 on page 8. |
|
Comments 26: paragraph 3.4, line 10, four groups at 12 h (not at 24 h too, please check in figure) |
|
Response 26: Thank you for your question, we checked Figure 4 A and there were no significant differences between the four treatment groups at 12 and 24 hours. |
|
Comments 27: paragraph 3.5, line 11, SOD mRNA expression levels at 6 and 24 h (not at four sampling times); also at line 19, CAT mRNA expression at 6 and 24 h (not at the four sampking times) |
|
Response 27: Thank you for pointing this out. We have made changes. “There were significant differences in SOD mRNA expression levels at 6 and 24 h for the three hypoxia stress groups,” “There were significant differences in CAT mRNA expression at 6 and 24 h for the three hypoxia stress groups,” Please see lines 468 on page 11, lines 481 on page 12. |
|
Comments 28: I suggest to uniform the scale of y-axis of Figures 4 and 5 (graph C), to make comparison of juveniles and subadults easier |
|
Response 28: Thank you for pointing this out. We have made changes. Please see lines 446-447 on page 11 and 504-505 on page 12. |
|
Comments 29: pag. 13, last line, juvenile crayfish were smaller (I suppose is subadult crayfish, please check) |
|
Response 29: Thank you for pointing this out. We have made a change. “while the fluctuations in LD content and LDH activity in the hepatopancreas of subadult crayfish were weaker than that of juvenile crayfish” Please see page 14 line 559-561. |
|
Comments 30: pag. 14, line 4, in the hepatopnacreas of juvenile shrimp compared to subadult shrimp (please check). Under hypoxia stress (instead of In the face of.... |
|
Response 30: Thank you for pointing this out. We have revised it. “Under hypoxia stress, juvenile crayfish will rapidly increase the anaerobic respiration rate in a short time (6 h) to generate energy and maintain body balance,” Please see lines 560, 567-568 on page 14. |
|
4. Response to Comments on the Quality of English Language |
|
Point 1: Minor editing of English language required |
|
Response 1: Thank you for your suggestion, the language has been checked and modified. |